# Regulation of Follicular Development in Chickens: *WIF1* Modulates Granulosa Cell Proliferation and Progesterone Synthesis via Wnt/β-Catenin Signaling Pathway

**DOI:** 10.3390/ijms25031788

**Published:** 2024-02-01

**Authors:** Ruixue Nie, Wenhui Zhang, Haoyu Tian, Junying Li, Yao Ling, Bo Zhang, Hao Zhang, Changxin Wu

**Affiliations:** State Key Laboratory of Animal Biotech Breeding, Beijing Key Laboratory for Animal Genetic Improvement, College of Animal Science and Technology, China Agricultural University, Beijing 100193, China; s20193040548@cau.edu.cn (R.N.); b20213040327@cau.edu.cn (W.Z.); 18286685342@163.com (H.T.); lijunying@cau.edu.cn (J.L.); lingzi@cau.edu.cn (Y.L.); chxwu@cau.edu.cn (C.W.)

**Keywords:** *WIF1*, steroid, *ESR2*, granulosa cell, follicle development, poultry

## Abstract

Proliferation, apoptosis, and steroid hormone secretion by granulosa cells (GCs) and theca cells (TCs) are essential for maintaining the fate of chicken follicles. Our previous study showed that the Wnt inhibitor factor 1 (*WIF1*) plays a role in follicle selection. However, the significance of *WIF1* in GC- and TC-associated follicular development was not explicitly investigated. This study found that *WIF1* expression was strongly downregulated during follicle selection (*p* < 0.05) and was significantly higher in GCs than in TCs (*p* < 0.05). *WIF1* inhibits proliferation and promotes apoptosis in GCs. Additionally, it promotes progesterone secretion in prehierarchal GCs (pre-GCs, 1.16 ± 0.05 ng/mg vs. 1.58 ng/mg ± 0.12, *p* < 0.05) and hierarchal GCs (hie-GCs, 395.00 ng/mg ± 34.73 vs. 527.77 ng/mg ± 27.19, *p* < 0.05) with the participation of the follicle-stimulating hormone (FSH). *WIF1* affected canonical Wnt pathways and phosphorylated β-catenin expression in GCs. Furthermore, 604 upregulated differentially expressed genes (DEGs) and 360 downregulated DEGs in *WIF1*-overexpressed GCs were found through RNA-seq analysis (criteria: |$\log_{2}\left(FoldChange\right)$| > 1 and *p*_adj < 0.05). Cytokine–cytokine receptor interaction and the steroid hormone biosynthesis pathway were identified. In addition, the transcript of estrogen receptor 2 (*ESR2*) increased significantly ($\log_{2}\left(FoldChange\right)$ = 1.27, *p*_adj < 0.05). Furthermore, we found that *WIF1* regulated progesterone synthesis by upregulating *ESR2* expression in GCs. Additionally, *WIF1* suppressed proliferation and promoted apoptosis in TCs. Taken together, these results reveal that *WIF1* stimulates follicle development by promoting GC differentiation and progesterone synthesis, which provides an insight into the molecular mechanism of follicle selection and egg-laying performance in poultry.

## 1. Introduction

The avian ovary is an important reproductive organ responsible for regulating the development and maturation of follicles and hormone secretion and, ultimately, influencing reproductive performance [1]. The follicle is the fundamental development unit of the ovary, where each oocyte is covered by the surrounding somatic granulosa cell (GC) layer and theca cell (TC) layer. Chicken ovary follicles can be categorized as follows: quiescent primordial follicles, slowly growing but easily atresic prehierarchal follicles, and well-ordered hierarchal follicles [2,3]. The process of recruiting into the cohort of hierarchal follicles in the ovaries typically observed in small yellow follicles (SYFs, 6–8 mm in diameter) is referred to as follicle selection [4]. The normal process of follicle selection is crucial for hens to achieve regular ovulation and sustain a high level of egg-laying performance.

Follicle selection is a precisely orchestrated process controlled by many regulatory molecules. The growth differentiation factor 9 (*GDF9*) and bone morphogenetic protein 15 (*BMP15*) genes from the TGF-β family are specifically expressed in oocytes and play a key role in the female fertility of birds [5]. The peptide growth factors, including the epidermal growth factor (EGF), likely act as a paracrine signal to regulate the proliferation of GCs during ovarian follicle development [6]. The excessive GC apoptosis and degeneration may result in follicular atresia [7]. The follicle-stimulating hormone (FSH), secreted by the pituitary gland, also participates in the regulation of follicle selection [8]. The accumulation of cyclic adenosine monophosphate (cAMP) induced by the FSH receptor (FSHR) is involved in the differentiation of GCs from undifferentiation status in prehierarchal follicles [1]. One of the most notable changes in differentiated GCs is the increased ability to synthesize steroid hormones, and the key enzymes involved in steroidogenesis are the steroidogenic acute regulatory protein (STAR), the cytochrome P450 family 11 subfamily A member 1 (CYP11A1), and 3 β-hydroxysteroid dehydrogenases (3β-HSDs) [9,10]. These observations suggest that cell proliferation, apoptosis, and steroid hormone synthesis and secretion can influence the fate of follicles. Although there are significant differences in GCs before and after follicle selection, the key regulatory factors that modulate GC differentiation from prehierarchal to hierarchal follicles have been uncovered. Additionally, the effects of TCs on follicular development have not been sufficiently investigated as the main membrane component of follicles.

Our previous study showed the mRNA expression profile of chicken follicles during selective development [11], and we identified the Wnt inhibitory factor 1 (*WIF1*) to be involved in follicle selection. The *WIF1* gene was originally identified in the human retina as a blocker of the Wnt signaling pathway [12]. This is well known for regulating human diseases, such as cancers of the prostate, breast, lung, and bladder [13,14]. Hypermethylation of the *WIF1* promoter is associated with various types of cancer, such as lung cancer and psoriasis [15,16]. However, the functions of *WIF1* in follicle development (especially in agricultural animals) remain unclear.

This study explored *WIF1* expression characteristics and biological functions, including proliferation, apoptosis, and steroid hormone secretion in chicken GCs and TCs, and the regulatory mechanisms of *WIF1* in GCs. To the best of our knowledge, this is the first study to elucidate the function and regulatory mechanism of *WIF1* during follicle selection in chickens, providing novel insights into the regulatory mechanism of follicle development and egg production in poultry.

## 2. Results

### 2.1. Expression Characteristics of WIF1 in Chickens

The expression of *WIF1* in prehierarchal follicles (large white follicle, LWFs or SYFs) was significantly higher than that in hierarchical follicles (F5s) (*p* < 0.05), which was consistent with our previous transcriptomic results (Figure 1A) [11]. Additionally, *WIF1* expression was significantly lower in LWFs and SYFs at the peak period (30 weeks old) of the laying cycle than at the late period (60 weeks old) (*p* < 0.05), but not statistically significant in F5s (*p* > 0.05, Figure 1A). *WIF1* expression showed dynamic changes in follicles during different developmental stages in chicken ovaries, and the *WIF1* expression level was highest in LWFs, gradually declined until F2s, and slightly increased in F1s (Figure 1B). The *WIF1* levels were significantly higher in GCs than in TCs throughout the follicle development process (*p* < 0.05, Figure 1C). Additionally, the mRNA and protein levels of *WIF1* in GCs were consistent with those in follicle tissues (*p* < 0.05, Figure 1C–E). *WIF1* was predominantly expressed in the ovarian stroma and lungs (*p* < 0.05; Figure 1F). These results suggest that *WIF1* plays a regulatory role in follicle development, specifically through its involvement in GCs.

### 2.2. WIF1 Inhibits Proliferation of Chicken GCs from Prehierarchal Follicles

The *WIF1* overexpression vector (pcDNA3.1-WIF1) or siRNA (siRNA-WIF1) was transfected into GCs to investigate the effect of *WIF1* on proliferation. Overexpression and knockdown efficiencies were detected using quantitative real-time PCR (qRT-PCR) and Western blotting. The mRNA and protein expression levels of *WIF1* in GCs significantly increased after transfection with pcDNA3.1-WIF1 (*p* < 0.001 and *p* < 0.01, respectively; Figure 2A–C). Three siRNA fragments were synthesized and transfected into GCs for knockdown; siRNA3-WIF1 significantly reduced the mRNA and protein expression levels of *WIF1* (*p* < 0.001 or *p* < 0.05, Figure 2D–F) and was selected for subsequent experiments. 5-Ethynyl-2′-deoxyuridine (EdU) staining demonstrated that the proliferation rate of *WIF1* overexpression in GCs was significantly lower than that of control cells (*p* < 0.01, Figure 2G,H). *WIF1* overexpression markedly reduced the absorbance of GCs after cell counting kit-8 (CCK-8) treatment (*p* < 0.001; Figure 2I). Moreover, qRT-PCR showed that *WIF1* overexpression significantly inhibited the mRNA expression of cell cycle-related genes, such as cyclin-dependent kinase 2 (*CDK2*), cyclin A1 (*CCNA1*), and cyclin A2 (*CCNA2*) (*p* < 0.05, Figure 2J). Conversely, *WIF1* silencing significantly increased the number of EdU-positive cells (*p* < 0.01, Figure 2K,L) and the absorbance values (*p* < 0.01, Figure 2M). *WIF1* knockdown significantly enhanced the expression of *CDK2*, *CCNA1*, and *CCNA2* (*p* < 0.05; Figure 2N). Taken together, these results suggest that *WIF1* inhibits the proliferation of chicken GCs.

### 2.3. WIF1 Promotes the Apoptosis of Chicken Prehierarchal GCs

We measured the rate of apoptosis using flow cytometry to determine the effect of *WIF1* on GC apoptosis. GCs positively stained with Annexin V-FITC and PI are considered apoptotic. The overexpression of *WIF1* significantly promoted the apoptosis of GCs (*p* < 0.05; Figure 3A,B). *WIF1* knockdown significantly decreased the rate of apoptosis (*p* < 0.05; Figure 3C,D). Taken together, these data suggest that *WIF1* promotes apoptosis in chicken ovarian GCs.

### 2.4. WIF1 Promotes the Secretion of Progesterone with the Participation of FSH in Chicken GCs

Chicken GCs respond to FSH and produce progesterone, which is a marker of GC differentiation [2]. Granulosa cells from prehierarchal follicles (pre-GCs) and hierarchal follicles (hie-GCs) were treated with different FSH concentrations (0, 10, 50, and 100 ng/mL) for 3 h or 24 h. Progesterone concentrations in the medium of pre-GCs did not show a significant increase (*p* > 0.05) after a 3 h treatment with FSH, whereas there was a significant dose-dependent increase in progesterone concentration in hie-GCs (*p* < 0.001, Appendix A). After 24 h, treatment with 50 and 100 ng/mL FSH resulted in a significant increase in the progesterone concentration in the medium of pre-GCs (*p* < 0.05, Appendix A). Moreover, the concentration of progesterone continued to increase with increasing concentrations of FSH in hie-GCs (*p* < 0.001, Appendix A). Based on these results, a significant increase in progesterone concentration was observed in pre-GCs and hie-GCs after 24 h of treatment with 50 ng/mL FSH; this concentration and duration were chosen for further experiments.

When FSH was not supplemented, the qRT-PCR results in pre-GCs indicated that *WIF1* overexpression did not have a significant stimulatory effect on the expression of progesterone synthesis-related genes, including *STAR*, *CYP11A1*, and *HSD3B1* (*p* > 0.05), whereas it significantly promoted the mRNA expression of *FSHR* (*p* < 0.01, Figure 4A–D). There was a significant increase in the expression levels of *STAR*, *HSD3B1*, and *FSHR* when pre-GCs were treated with a combination of FSH and *WIF1* (*p* < 0.05, Figure 4A–D). However, *WIF1* knockdown did not significantly affect the expression of genes involved in progesterone synthesis (*p* > 0.05; Appendix A). In hie-GCs (without FSH supplementation), qRT-PCR results demonstrated that *WIF1* overexpression significantly promoted the mRNA expression levels of *HSD3B1* and *FSHR* (*p* < 0.01, Figure 4E–H). Furthermore, there was a significant increase in the expression levels of *STAR*, *CYP11A1*, *HSD3B1*, and *FSHR* when hie-GCs were treated with a combination of FSH and *WIF1* (*p* < 0.05, Figure 4E–H). In contrast, *WIF1* silencing significantly inhibited the mRNA expression of *STAR* (*p* < 0.05; Appendix A).

The Western blotting results indicate that *WIF1* overexpression significantly promoted STAR protein expression in pre-GCs in the presence of FSH (*p* < 0.05, Figure 4I,J) and hie-GCs (*p* < 0.05, Figure 4K,L). Simultaneous treatment with FSH and *WIF1* overexpression significantly promoted progesterone production in the pre-GC and hie-GC media (*p* < 0.05; Figure 4M,O). Conversely, progesterone production was significantly inhibited after *WIF1* knockdown (*p* < 0.05; Figure 4N,P). In summary, *WIF1* potentiated the FSH-mediated induction of progesterone synthesis in chicken GCs.

### 2.5. WIF1 Inhibits Wnt Signaling Activity in Chicken GCs

Top/Fop flash luciferase reporter assays revealed that Wnt signaling activity in GCs gradually increased with development and significantly decreased in preovulation follicles (F1s) (*p* < 0.05, Figure 5A). *WIF1* overexpression markedly attenuated the transcriptional activation of TCFs/LEFs in SYF-GCs and F5-GCs (*p* < 0.05; Figure 5B), whereas *WIF1* knockdown increased transcriptional activation (*p* < 0.05; Figure 5C). Western blot assays showed that *WIF1* overexpression had no effect on β-catenin (*p* > 0.05) but upregulated the phosphorylation level of β-catenin in GCs (*p* < 0.05, Figure 5D,E). Meanwhile, the mRNA levels of *TCF1*, which is an element of Wnt/β-catenin signaling, were decreased after *WIF1* overexpression (*p* < 0.001, Figure 5F). Collectively, these data suggest that *WIF1* overexpression inhibits the Wnt/β-catenin signaling pathway in GCs.

### 2.6. WIF1 Stimulates Metabolic Activity of Chicken Prehierarchal GCs

Global transcriptomic analysis was performed using RNA-seq with *WIF1*-overexpressed and control pre-GCs to explore the regulatory mechanism of *WIF1* in GCs. The quality control and the analysis information for the raw data are provided in Appendix A; the differentially expressed genes (DEGs) include 604 upregulated and 360 downregulated genes (criteria: |$\log_{2}\left(FoldChange\right)$| > 1 and *p*_adj < 0.05; Figure 6A). The DEGs were mainly involved in cell differentiation, regulation of cellular processes, and positive metabolic processes compared to the control group (pcDNA3.1) by GO enrichment analysis (Figure 6B, Appendix A). Kyoto Encyclopedia of Genes and Genomes (KEGG) enrichment analyses of DEGs and gene set enrichment analysis (GSEA) were performed for all genes (Figure 6C, Appendix A); KEGG analysis identified several significantly associated metabolic pathways, including glycine, serine, and threonine metabolism (*p* = 0.012); cysteine and methionine metabolism (*p* = 0.007); and amino acid biosynthesis (*p* = 0.007) (Appendix A). Additionally, the genes enriched in these metabolism-related pathways were upregulated in the *WIF1*-overexpressed group (pcDNA3.1-WIF1) (Figure 6D).

Eleven pathways were present in the KEGG (*p* < 0.05) and GSEA analyses, and some of these pathways aligned with the GC phenotypes mentioned earlier (Figure 6C). Among these 11 pathways, the cytokine–cytokine receptor interaction pathway (Figure 6E) and the cell cycle pathway were significant (Figure 6F), since most genes in these two pathways exhibited upregulation and downregulation patterns (Figure 6I). This indicates that these pathways were potentially relevant to the observed cellular characteristics. The calcium signaling pathway (Figure 6G) and the toll–like receptor signaling pathway (Figure 6H) were identified using KEGG and GSEA, and the genes enriched in these pathways were highly expressed in the *WIF1* overexpression group (Figure 6I). Accordingly, metabolism-related pathways and cytokine–cytokine receptor interaction pathways may be involved in enhanced progesterone synthesis induced by *WIF1* overexpression. We hypothesized that *WIF1* may affect the calcium and toll–like receptor signaling pathways that regulate chicken GCs.

### 2.7. WIF1 Promotes Progesterone Synthesis through Upregulation of ESR2 in GCs

Estrogen receptor 2 (*ESR2*) was significantly upregulated among DEGs that responded to *WIF1* overexpression (Figure 7A). *ESR2* is an estrogen receptor that plays a critical role in folliculogenesis [17]. The depletion of both estrogen receptors results in infertility in mice [18]. Accordingly, we hypothesized that *WIF1* may regulate progesterone synthesis through *ESR2* in GCs. The qRT-PCR results show that *ESR2* expression was remarkably promoted after *WIF1* overexpression (Figure 7B). siRNA1-ESR2 significantly reduced the mRNA expression levels of *ESR2* (*p* < 0.05, Figure 7C) and was selected for subsequent experiments. To test our hypothesis, GCs were co-transfected with pcDNA3.1-WIF1, siRNA-NC, or siRNA-ESR2. siRNA1-ESR2 inhibited the increase in *FSHR*, *STAR*, and *HSD3B1* induced by *WIF1*, but not *CYP11A1* (*p* < 0.05, Figure 7D–G). The protein abundance of STAR declined by siRNA1-ESR2 on the basis of being increased by *WIF1* (*p* < 0.05, Figure 7H,I). Subsequently, the fluorescence signal of *FSHR* was enhanced in GCs transfected with pcDNA3.1-WIF1 and slightly diminished after co-transfection with pcDNA3.1-WIF1 and siRNA1-ESR2 (Figure 7J). Moreover, progesterone concentration in the GC medium declined after co-transfection with pcDNA3.1-WIF1 and siRNA1-ESR2 (Figure 7K). In summary, these results indicate that *WIF1* regulates steroidogenesis-related genes by upregulating *ESR2* and promoting progesterone synthesis in GCs.

### 2.8. WIF1 Suppresses TC Proliferation and Promotes Apoptosis

Quantitative RT-PCR analysis showed that *WIF1* expression increased by approximately 6,000 in TCs 24 h after transfection with the *WIF1* overexpression vector (*p* < 0.001; Figure 8A). EdU staining demonstrated that the proliferation rate of *WIF1*-overexpressed cells was significantly lower than that of the control cells (*p* < 0.01; Figure 8B). In addition, the TUNEL assay showed a significant increase in TC apoptosis after *WIF1* overexpression (*p* < 0.05; Figure 8C). However, there was no significant difference in the mRNA expression of *CYP17A1*, *CYP19A1*, *HSD17B1*, or the secretion of estradiol in TCs after *WIF1* overexpression (*p* > 0.05, Figure 8D,E).

## 3. Discussion

The regulation of follicular development is a complex and multicellular process in all vertebrates. *WIF1* is a secreted Wnt signaling antagonist that is highly expressed and largely conserved from teleosts to humans [19]. In mice, *WIF1* expression is retained in the heart and lung and is expressed in the brain and eye [12]. Mammalian *WIF1* is associated with lung development [20]. *WIF1* interacts with the Olfactomedin protein to regulate the intraocular pressure of the eye [21,22,23]. In chickens, Chen et al. have identified *WIF1* as significantly downregulated in F6 follicles compared with its expression in SYFs at both the mRNA and protein levels [24]. However, there is a lack of functional studies of the *WIF1* gene in agricultural animals, especially those related to follicular development. To the best of our knowledge, this is the first study demonstrating that *WIF1* expression declined during chicken follicle selection and shows a slight increase in F1s at the mRNA and protein levels. Notably, *WIF1* is mainly expressed in GCs rather than in TCs. Based on these expression characteristics in chicken follicles, *WIF1* may play an important role in follicle development through GCs, especially follicle selection.

Our in vitro studies demonstrated that *WIF1* overexpression decreased cell viability (detected by the CCK8 assay) and the EdU-positive cell rate, thereby inhibiting follicular GC proliferation. *CDK2*, *CCNA1*, and *CCNA2* are proliferation markers that promote proliferation or follicular development [25,26,27]. Furthermore, we detected the expression levels of several key genes associated with proliferation. The qRT-PCR results show that the expression levels of the *CDK2*, *CCNA1*, and *CCNA2* proliferation markers significantly decreased with *WIF1* overexpression (Figure 2J), and these were significantly increased by *WIF1* knockdown (Figure 2N). This inhibition of cell proliferation is consistent with previous research on cancer cell lines, such as glioblastoma cell lines, cervical cancer cells, and basal cell carcinoma-related keratinocytes [14,28,29]. Granulosa cell apoptosis can impact follicular development in mammals [30]. In chickens, follicular atresia is caused by GC apoptosis, which occurs at any stage of follicle development, especially in the prehierarchal follicles [31]. Through flow cytometry, we found that *WIF1* promoted apoptosis of chicken GCs (Figure 3). In contrast, *WIF1* upregulation suppresses apoptosis in osteoarthritic chondrocytes [32]. Consistent with our in vitro results, *WIF1* gene transfer induced significant apoptosis in cervical cancer cells in vitro [29].

Progesterone is a member of the steroid hormone family, and steroidogenesis is the process wherein cholesterol is transformed into biologically active steroid hormones under the control of multiple enzymes [33]. In ovarian follicles, steroidogenesis is mainly confined to GCs and TCs, with the regulation of FSH and luteinizing hormone (LH) via the cAMP signaling pathway [34]. In chicken GCs, steroidogenic capacity is directly associated with the transition of prehierarchal follicles to hierarchal follicles (follicle selection), which leads to follicle growth and maturation. In particular, cholesterol in GCs is transported to the mitochondria mediated by STAR and is subsequently converted into progesterone through the sequential catalysis of CYP11A1 and HSD3B1 [10]. In our study, the mRNA levels of *STAR*, *CYP11A1*, and *HSD3B1* were significantly increased by pcDNA3.1-WIF1, and the protein level of STAR was upregulated after *WIF1* overexpression in hie-GCs treated with a combination of FSH (Figure 4). However, these results are uncommon. Other studies show that the proliferation and progesterone-synthesizing capacity of GCs showed the same trend; that is, an increased proliferative capacity was often accompanied by an increased progesterone-synthesizing capacity. Wnt4 stimulated the proliferation of follicular GCs and increased the expression of *STAR* and *CYP11A1* mRNA in prehierarchal and hierarchal follicles in chickens [4]. LncRNA is an inhibitory factor of follicular development (IFFD) that arrests follicle development by inhibiting the proliferation and estrogen secretion of GCs [35]. Meanwhile, isorhamnetin can inhibit the secretion of progesterone and promote the proliferation of GCs via the PI3K/Akt signaling pathway in porcine [36]. In summary, we believe that follicular development is a dynamic, balanced process regulated by multiple factors. High *WIF1* expression in prehierarchical follicles can prevent premature development by inhibiting proliferation and promoting the apoptosis of GCs. After follicular GCs receive FSH signaling, *WIF1* promotes *FSHR* expression (Figure 4D) in GCs to synthesize large amounts of progesterone, which is converted to differentiated hierarchical follicles. After follicle selection, even a small amount of *WIF1* can sustain the requirement for large amounts of progesterone synthesis because GCs are already in a differentiated state.

The Wnt/β-catenin signaling pathway (also named the canonical Wnt pathway) is vital to follicular development [37,38]. As a secreted Wnt antagonist, WIF1 can directly bind to Wnt molecules to prevent the Wnt/β-catenin signaling pathway [19]. This prevention also influences the downstream target genes of c-Myc and Cyclin D1 of the Wnt/β-catenin signaling pathway [39,40,41]. Interestingly, Top/Fop flash reporter assays showed that the Wnt/β-catenin signaling activity in GCs gradually increased during follicle development and significantly declined in F1s. Furthermore, *WIF1* overexpression upregulated phosphorylated β-catenin in GCs, which was detected by Western blotting, and inhibited Wnt/β-catenin signaling activity (Figure 5). Our study suggested that *WIF1* regulated chicken GC development by suppressing the Wnt/β-catenin signaling pathway.

Accumulating evidence indicates that follicle development is modulated by several molecules and signaling pathways [42,43]. To characterize the molecular features of *WIF1*’s regulatory role in GCs, RNA-seq was performed on GCs overexpressing *WIF1*. *WIF1* overexpression promoted the expression of genes enriched in metabolic processes, including glycine, serine, threonine, cysteine, and methionine metabolism; amino acid biosynthesis; and steroid hormone biosynthesis (Appendix A). These findings suggest that *WIF1* promotes metabolic activity in GCs. Interestingly, previous research shows that *WIF1* is necessary for achieving periportal zone-specific characteristics, such as the enhancement of gluconeogenesis capacities [44]. Vassallo et al. observed that the *WIF1* expression vector downregulated intracellular Ca^2+^ in LN-229 cells [45]. Hence, these results suggest that *WIF1* can alter the metabolic state of cells by regulating intracellular ion homeostasis. In this study, the calcium and toll-like receptor signaling pathways were identified by KEGG and GSEA enrichment, and the genes enriched in these pathways were highly expressed in the *WIF1* overexpression group (Figure 6). Based on our experimental data, we propose that *WIF1* influences intracellular metabolic activities through the calcium signaling pathway in chicken follicular GCs. This, in turn, promotes whole-follicle maturation.

In chickens, the oocyte is covered by surrounding somatic cells (GCs and TCs) to form the functional unit in female reproduction [46]. Also, GCs have been studied much more than TCs since they are major components of the ovarian follicle. Our data show that *WIF1* overexpression decreased the proliferation rate and significantly increased the apoptosis of TCs (Figure 7), which was consistent with the phenotype of GCs. However, there was no significant difference in estradiol secretion by TCs following *WIF1* overexpression (Figure 7). According to the classical three-cell model in avian species, progesterone is produced by GCs and is subsequently converted into estradiol by TCs [46]. This may be because culturing TCs alone leaves them without sufficient progesterone as a raw material for estradiol synthesis, ultimately resulting in no difference in estradiol concentration. In summary, these results demonstrate that *WIF1* plays an important role in follicular development by promoting progesterone synthesis of GCs, which may provide an insight into the regulatory mechanisms of follicle selection and egg-laying performance in poultry.

## 4. Materials and Methods

### 4.1. Ethics Statement and Experimental Animals

All animal experimental protocols conducted in this study were approved by the Animal Care and Use Committee of China Agricultural University and performed in compliance with the National Research Council’s Guide for the Care and Use of Laboratory Animals (AW80203202-1-1). All animal experiments were approved by the Experimental Chicken Farm of China Agricultural University (Beijing, China).

Yellow-bearded chickens were housed in separate cages with the same environmental conditions and a daily light period of 14 h. All chickens had free access to water and feed. After euthanizing the chickens (at the age of 30 weeks), stroma, heart, liver, lung, gizzard, kidney, breast muscle, and abdominal fat were collected and preserved at −80 °C (*n =* 5). Follicles from laying chickens (during the egg-laying period, at the age of 30–50 weeks) were categorized into two groups: prehierarchal and hierarchal. The prehierarchal follicles were further classified into different categories based on their diameters: small white follicles (SWFs, diameter 1–3.9 mm), large white follicles (LWFs, 4–5.9 mm), and small yellow follicles (SYFs, diameter 6–8 mm). There are generally five hierarchal follicles in the hen ovary: the largest is F1, the second largest is F2, and so on.

### 4.2. Cell Isolation and Culture

Different stages of GCs were isolated from the ovary follicles of laying hens based on experimental requirements. After removing connective tissue from the follicle surface, the yolks of the follicles were carefully removed in phosphate-buffered saline (PBS). The GC (the layer adjacent to the egg yolk) and TC (outer-side layer of the GC) layers were isolated using a stereomicroscope (SZ2-ILST; Olympus, Shibuya, Japan). Subsequently, these were subjected to enzymatic digestion by collagenase type II (Sigma Aldrich, Inc., St. Louis, MO, USA) at 37 °C; the GC and TC layers were digested for 5 min and 30 min, respectively. The cell suspension containing the GC or TC layer was filtered using cell strainers (Biosharp, Hefei, China) with a pore size of 50 μm, and the filtration process was repeated 3 to 5 times. The cells were maintained in a basal medium consisting of Dulbecco’s modified Eagle medium (DMEM) (Gibco, Gaithersburg, MD, USA) with 10% fetal bovine serum (FBS) (Gibco) and 1% penicillin–streptomycin (Gibco) in a 37 °C, 5% CO_2_ humidified atmosphere incubator.

### 4.3. Plasmid Construction, RNA Oligonucleotide Synthesis, and Cell Transfection

To construct the *WIF1* overexpression vector, the complete coding sequences of the *WIF1* gene along with the EcoRI and BamHI restriction enzyme sites were cloned into a pcDNA3.1-3×flag vector with primers (forward primer: 5′-tagcgtttaaacttaagcttATGGCCGCGGCGGGGCGGCTGT-3′; reverse primer: 5′-gctggatatctgcagaattcCCAGATATAATTGGATTCAGGT-3′). This was performed using Q5^®^ High-Fidelity 2X Master Mix (M0492S, New England Biolabs) and the Trelief™ SoSoo Cloning Kit (TSINGKE, Beijing, China). The modified vector was named pcDNA3.1-WIF1. Plasmid DNA was subsequently extracted and purified using an EndoFree Maxi Plasmid Kit (TIANGEN, Beijing, China) according to the manufacturer’s instructions.

Small interfering RNA (siRNA) oligonucleotides were designed and synthesized by GenePharma Co., Ltd. (Shanghai, China) to specifically knockdown the *WIF1* gene, including siRNA1-WIF1 (sense: 5′-GCAUCCACGCCAUGAACUUTT-3′; antisense: 5′-AAGUUCAUGGCGUGGAUGCTT-3′), siRNA2-WIF1 (sense: 5′-GGCUGAUCCAACUGUAAAUTT-3′; antisense: 5′-AUUUACAGUUGGAUCAGCCTT-3′), and siRNA3-WIF1 (sense: 5′-GGGAUUCGAAGGAGACCAATT-3′; antisense: 5′-UUGGUCUCCUUCGAAUCCCTT-3′). siRNA3-WIF1 was selected for subsequent experiments with the highest knockdown efficiency.

Briefly, cell transfection experiments were performed using Lipofectamine^®^ 3000 Transfection Reagent (Invitrogen, Carlsbad, CA, USA) and Opti-MEM medium (Gibco) according to the manufacturer’s instructions.

### 4.4. Quantitative Real-Time PCR (qRT-PCR)

qRT-PCR is a method of detecting the relative expression quantification of a gene. Tissues and cells were lysed using TRIzol reagent (TIANGEN) for total RNA extraction. For each sample, 2 μg of total RNA was reverse-transcribed to generate cDNA using a FastKing RT Kit (TIANGEN). qRT-PCR was performed in a 20 μL reaction volume containing 10 μL of 2×Universal SYBR Green Fast qPCR Mix (ABclonal, Wuhan, China), 8 μL of RNase-free water, 0.5 μL each of forward and reverse primers (10 μmol/L), and 1 μL of cDNA (approximately 300 ng). qRT-PCR was performed using a CFX96 Real-Time System (Bio-Rad, Hercules, CA, USA). The expression levels of the coding genes were normalized to those of β-actin. Primer sequences were designed using the Primer-BLAST tool in the National Center for Biotechnology Information (NCBI) database and are listed in Appendix A.

### 4.5. Western Blotting

Total protein was extracted using RIPA lysis buffer with a protease inhibitor cocktail (Beyotime Biotechnology, Shanghai, China), and the concentration of each sample was determined using the bicinchoninic acid (BCA) Protein Assay Kit (Beyotime). A total of 15 microliters of protein was mixed with 5 μL of sodium dodecyl sulfate polyacrylamide gel electrophoresis (SDS-PAGE) loading buffer (Beijing Solarbio Science & Technology Co., Ltd., Beijing, China) and denatured at 95 °C for 5 min. The protein (20 μL) was separated by electrophoresis and transferred onto polyvinylidene fluoride (PVDF) membranes (Bio-Rad). The membranes were blocked in QuickBlock™ Blocking Buffer (Beyotime) for 20 min at 25 ± 2 °C and incubated with primary antibody (Appendix A) and secondary antibody (Solarbio, diluted ratio at 1:5000) solutions in turn. The protein band was visualized using the Super ECL Prime Kit (US Everbright^®^ Inc., Suzhou, China). GAPDH was used as the reference protein, and the grayscale value of the band was quantified using ImageJ v2.0 software [47].

### 4.6. 5-Ethynyl-2′-Deoxyuridine (EdU) Assay

The EdU assay could accurately reflect the proliferation of cells. The EdU assay was performed according to the instructions of the Cell-Light EdU Apollo567 In Vitro Kit (RiboBio, Guangzhou, China). Cells were seeded in 48-well plates and transfected with plasmids or siRNAs. The cells were stained with EdU reagent for 2 h, and the cell nuclei were labeled with Hoechst (Beyotime). The fields were observed and photographed using an Echo Revolve fluorescence microscope (Echo Laboratories, San Diego, CA, USA). EdU-positive and total cells were counted using ImageJ v2.0 software [47].

### 4.7. Cell Counting Kit-8 (CCK 8) Assay

A widely used method for detecting cell activity, the cell counting kit-8 (Beyotime), was used to analyze cell proliferation. Granulosa cells were seeded in 96-well plates and cultured in basal medium. The CCK-8 solution was added to the wells at 6, 12, 24, and 48 h after transfection and incubated for 1 h. The absorbance was measured at 450 nm using a SpectraMax^®^ i3x Multi-Mode Microplate Reader (Molecular Devices Corporation, Sunnyvale, CA, USA).

### 4.8. Flow Cytometry

The apoptosis rate of GCs was evaluated using an Annexin V-FITC Apoptosis Detection Kit (Beyotime). Briefly, cells were cultured in 6-well plates and transfected for 24 h. The cells were collected and suspended in 1× buffer containing Annexin V-FITC and propidium iodide (PI). Cells were analyzed by flow cytometry using a BD FACSCalibur flow cytometer (BD Biosciences, San Diego, CA, USA), and the data were analyzed using FlowJo_v10.8.1 software. Flow cytometry was used to divide the cells into four quadrants: Q1 (mechanically injured cells), Q2 (late apoptotic cells), Q3 (early apoptotic cells), and Q4 (live cells).

### 4.9. Enzyme-Linked Immunosorbent Assay (ELISA)

The concentration of hormones in the supernatant of the cell culture medium was detected using a chicken progesterone/estradiol ELISA kit (Beijing SINO-UK Institute of Biological Technology, Beijing, China) according to the manufacturer’s instructions. The intracellular total protein content was assessed using a BCA Protein Quantification Kit (Beyotime) to normalize the progesterone hormone content.

### 4.10. Dual-Luciferase Reporter Assay

The luciferase reporter assay was performed with the Dual-Luciferase^®^ Reporter Assay System Kit (Promega, WI, USA) following the manufacturer’s instructions. The Top/Fop flash luciferase reporter system was used to determine the activity of Wnt signaling at different developmental stages in GCs. The pRL-TK Renilla plasmid and the indicated plasmids (Top flash or Fop flash) were co-transfected. After 24 h, luciferase and Renilla signals were measured on a SpectraMax^®^ i3x Multi-Mode Microplate Reader (Molecular Devices Corporation). Renilla luciferase activity was normalized to firefly luciferase activity.

### 4.11. RNA-seq and Bioinformatics Analysis

RNA-seq analysis for the WIF1-overexpressed GCs and controls was performed by Frasergen Information Co., Ltd. (Wuhan, China). Raw RNA-seq data were deposited in the National Center for Biotechnology Information (NCBI)’s Sequence Read Archive (SRA) database under the accession number PRJNA1025264. Clean reads were generated after removing adapters and low-quality reads from the raw data using fastp v0.20.1 [48]. The clean reads were aligned to the chicken reference genome (Gallus-gallus-7.0) using HISAT2 v2.2.1 [49]. The expression levels were normalized to fragments per kilobase of transcript per million mapped fragments (FPKMs) using the Cufflinks v2.2.1 software [50]. DEGs were identified using DESeq2 v1.32.0 [51]. Enrichment analyses, including Gene Ontology (GO), Kyoto Encyclopedia of Genes and Genomes (KEGG), and gene set enrichment analysis (GSEA) (criteria: |NES| > 1, *p*_value < 0.05, and false discovery rate (FDR) < 25%), were performed using R package clusterProfiler [52].

### 4.12. Co-Transfection of Plasmid DNA and Small Interfering RNA (siRNA)

The siRNAs of estrogen receptor 2 (ESR2) were designed and synthesized by GenePharma, including siRNA1-ESR2 (sense: 5′-GCAUUCUGCAGUCCUGCUATT-3′; antisense: 5′-UAGCAGGACUGCAGAAUGCTT-3′); siRNA2-ESR2 (sense: 5′-GUCAGUCAUCACUUCUGUATT-3′; antisense: 5′- UACAGAAGUGAUGACUGACTT-3′); and siRNA3-ESR2 (sense: 5′-GCUACGGAAAUGCUAUGAATT-3′; antisense: 5′-UUCAUAGCAUUUCCGUAGCTT-3′). siRNA1-ESR2 was selected for subsequent experiments with the highest knockdown efficiency. Co-transfection of plasmid and siRNA was performed using Lipofectamine^®^ 2000 Reagent (Invitrogen) by adding ~0.6 μg of siRNA per 1 μg of DNA.

### 4.13. Immunofluorescence Staining

Chicken GCs were washed with PBS, fixed with 4% paraformaldehyde for 30 min, and permeabilized with 0.5% Triton X-100 for 15 min. The GCs were supplemented with diluted goat serum blocking buffer (Solarbio, Cat# SL038) for 1 h, incubated with FSHR Rabbit Polyclonal antibody (Proteintech, Chicago, IL, USA, 22665-1-AP, 1:500) at 4 °C overnight, and incubated with fluorescently labeled secondary antibody (Proteintech, Alexa Flour 488, 1:1000) for 1 h at room temperature. Nuclei were stained with Hoechst stain (Beyotime) for 30 min. Images were visualized using a laser scanning confocal microscope (Nikon A1R; Tokyo, Japan) with the same parameters.

### 4.14. TdT-Mediated dUTP Nick-End Labeling (TUNEL) Assay

Apoptosis of chicken TCs was examined using a One-Step TUNEL Apoptosis Assay Kit (Beyotime) according to the manufacturer’s protocol. Initially, TCs were seeded in 24-well plates and incubated in 4% paraformaldehyde (Beyotime), followed by 1% Triton X-100 (Sigma). The cells were subsequently treated with a freshly prepared TUNEL reaction mixture. The ratio of TUNEL-positive cells was calculated based on double staining with TUNEL (green) and Hoechst (blue).

### 4.15. Statistical Analysis and Data Visualization

Data are expressed as means ± standard error (SE). Statistical analyses were conducted using *t*-tests or one-way analysis of variance (ANOVA) using SPSS v25 (SPSS Inc., Chicago, IL, USA). Lowercase letters a, b, c, d, and e indicate the level of significance. * *p* < 0.05, ** *p* < 0.01, and *** *p* < 0.001. ns, non-significant differences (*p* ≥ 0.05). Graphical visualizations were created using GraphPad Prism version 8.0 (GraphPad Software, San Diego, CA, USA), ggplot2 [53], TBtools v0.6673 [54], EVenn [55], and GSEA plot [56]. A schematic diagram was generated using Figdraw (www.figdraw.com).

## 5. Conclusions

In conclusion, our study demonstrates that *WIF1* is differently expressed during chicken follicle selection and regulates GC and TC functions such as cell proliferation, apoptosis, and hormone secretion through Wnt/β-catenin signaling pathways. To the best of our knowledge, this is the first study to report the function and regulatory mechanism of *WIF1* in chicken granulosa cells. These results elucidate a new molecular regulatory mechanism for follicle selection in chickens, which provides a new insight into potential applications in assisted reproduction and breeding technologies in poultry.

## Figures and Tables

**Figure 1 ijms-25-01788-f001:**
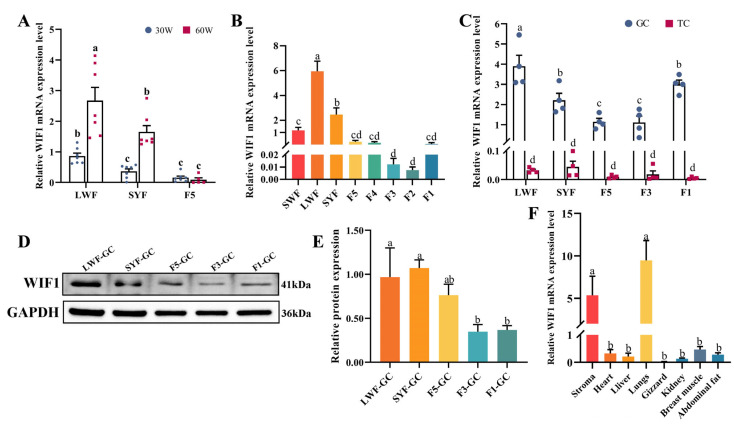
Expression characteristics of *WIF1* in chickens. (**A**) mRNA expression level of *WIF1* in chicken follicles was compared between the peak laying period (30 weeks old) and the late laying period (60 weeks old) (*n* = 7). (**B**) *WIF1* expression pattern in follicles during different development stages on the chicken ovary at 30 weeks old (*n* = 5). SWF, small white follicle; LWF, large white follicle; SYF, small yellow follicle; F5–F1 represent hierarchal follicles, which are sorted from smallest to largest in diameter. (**C**) *WIF1* expression level in granulosa cells (GCs) and theca cells (TCs) from several representative developmental stages (*n* = 4). (**D**,**E**) WIF1 protein expression level in GCs of several follicles detected by Western blotting (*n =* 3). The band intensities were measured by ImageJ and normalized to GAPDH. (**F**) mRNA expression level of *WIF1* in 8 different tissues in chickens (*n =* 5). Results are shown as the mean ± SEM. Differences between groups were determined by ANOVA. Different lowercase letters are used to indicate the level of significance of differences (*p* < 0.05); the same lowercase letters indicate non-significant differences (*p* > 0.05).

**Figure 2 ijms-25-01788-f002:**
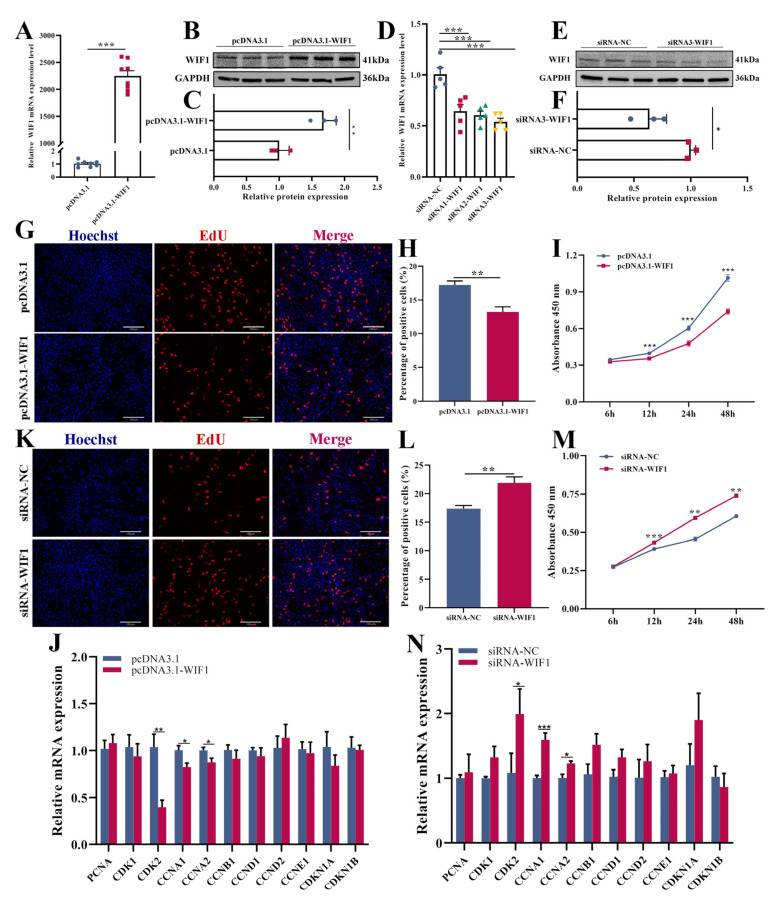
*WIF1* inhibits the proliferation of chicken prehierarchal granulosa cells (GC). mRNA (**A**) (*n =* 8) and protein (**B**,**C**) (*n =* 3) expression levels of *WIF1* after transfection with pcDNA3.1-WIF1 in GCs. mRNA (**D**) (*n =* 5) and protein (**E**,**F**) (*n =* 3) expression levels of *WIF1* after transfection with siRNA-WIF1 in GCs. (**G**) Proliferation of chicken GCs determined by 5-ethynyl-2′-deoxyuridine (EdU) after 24 h of transfection with pcDNA3.1-WIF1, scale bar: 130 μm. (**H**) Histogram showing the proportion of EdU-positive cells using ImageJ (*n =* 3). (**I**) Cell counting kit-8 (CCK-8) assay of GCs transfected with pcDNA3.1-WIF1 at 6, 12, 24, and 48 h post-transfection (*n =* 10). (**J**) Expression levels of cell cycle-related genes in GCs with *WIF1* overexpression (*n =* 6). (**K**) Proliferation of chicken GCs determined by 5-ethynyl-2′-deoxyuridine (EdU) after 24 h of transfection with siRNA3-WIF1, scale bar: 130 μm. (**L**) Histogram showing the proportion of EdU-positive cells using ImageJ (*n =* 3). (**M**) Cell counting kit-8 assay of GCs transfected with siRNA3-WIF1 at 6, 12, 24, and 48 h post-transfection (*n =* 10). (**N**) Expression levels of cell cycle-related genes in GCs with *WIF1* knockdown (*n =* 6). Results are shown as the mean ± SEM. * *p* < 0.05, *** p* < 0.01, and *** *p* < 0.001.

**Figure 3 ijms-25-01788-f003:**
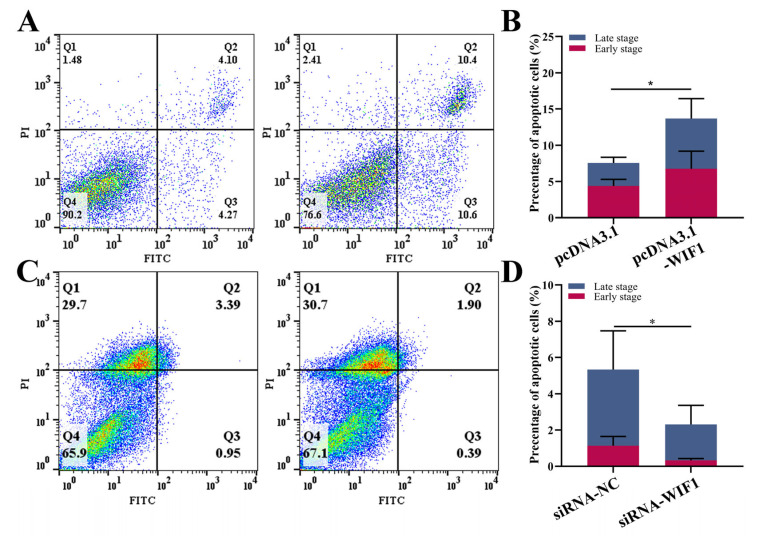
*WIF1* promotes the apoptosis of chicken prehierarchal GCs. (**A**,**B**) The apoptosis rates of GCs with *WIF1* overexpression were assessed by flow cytometry (*n =* 5). (**C**,**D**) The apoptosis rates of GCs with *WIF1* knockdown were assessed by flow cytometry (*n =* 5). * *p* < 0.05.

**Figure 4 ijms-25-01788-f004:**
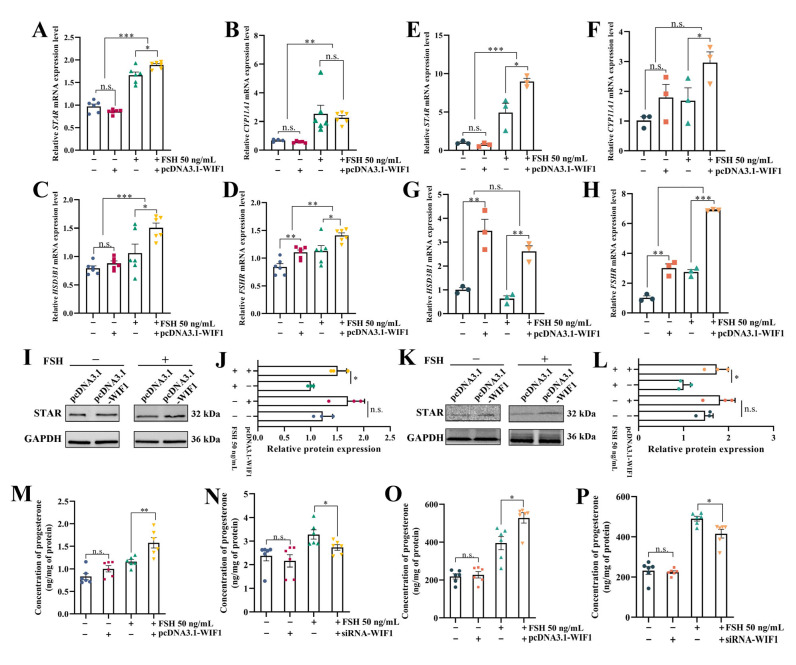
*WIF1* promotes the secretion of progesterone with the participation of FSH in chicken GCs. (**A**–**D**) mRNA expression levels of *STAR*, *CYP11A1*, *HSD3B1*, and *FSHR* in pre-GCs with *WIF1* overexpression (*n =* 6). (**E**–**H**) mRNA expression levels of *STAR*, *CYP11A1*, *HSD3B1*, and *FSHR* in hie-GCs with *WIF1* overexpression (*n =* 3). (**I**,**J**) Protein expression levels of STAR in pre-GCs with *WIF1* overexpression (*n =* 3). (**K**,**L**) Protein expression levels of STAR in hie-GCs with *WIF1* overexpression (*n =* 3). (**M**,**N**) Progesterone concentration in chicken pre-GCs with *WIF1* overexpression and inhibition was assessed by ELISA (*n =* 6). (**O**,**P**) Progesterone concentration in chicken hie-GCs with *WIF1* overexpression and inhibition was assessed by ELISA (*n =* 6). The intracellular total protein content was assessed to normalize the progesterone hormone content. Results are shown as the mean ± SEM. * *p* < 0.05, ** *p* < 0.01, *** *p* < 0.001, and n.s. *p* ≥ 0.05.

**Figure 5 ijms-25-01788-f005:**
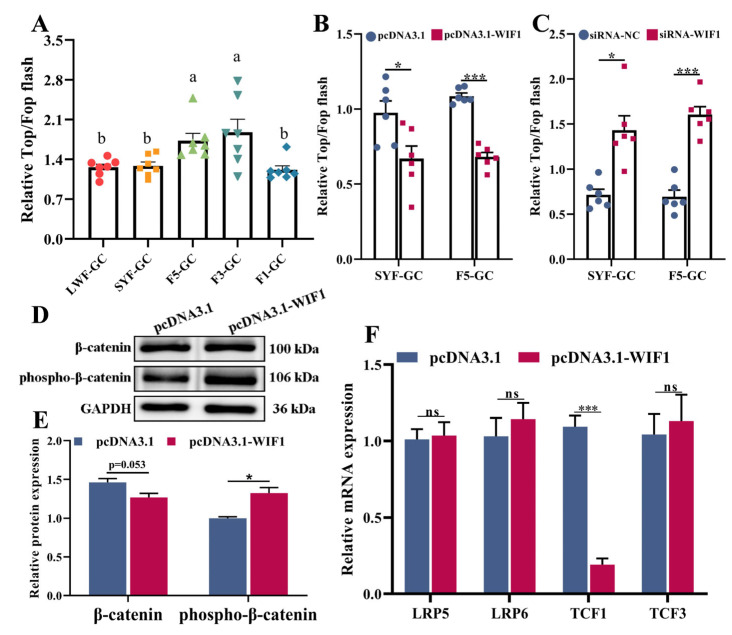
*WIF1* inhibits Wnt signaling activity in chicken GCs. (**A**) Luciferase reporter assays of Top/Fop transcriptional activity in GCs from different-development-stage follicles (*n =* 7). Different lowercase letters are used to indicate the level of significance of differences (*p* < 0.05); the same lowercase letters indicate non-significant differences (*p* > 0.05). (**B**) Luciferase reporter assays of Top/Fop transcriptional activity in GCs with *WIF1* overexpression (*n =* 6). (**C**) Luciferase reporter assays of Top/Fop transcriptional activity in GCs with *WIF1* inhibition (*n =* 6). (**D**,**E**) Western blotting analysis of β-catenin and phosphorylated β-catenin in GCs (*n =* 3). (**F**) Quantitative RT-PCR of the mRNA expression of some elements in Wnt/β-catenin signaling (*n =* 6). Results are shown as the mean ± SEM. * *p* < 0.05, *** *p* < 0.001, and ns *p* ≥ 0.05.

**Figure 6 ijms-25-01788-f006:**
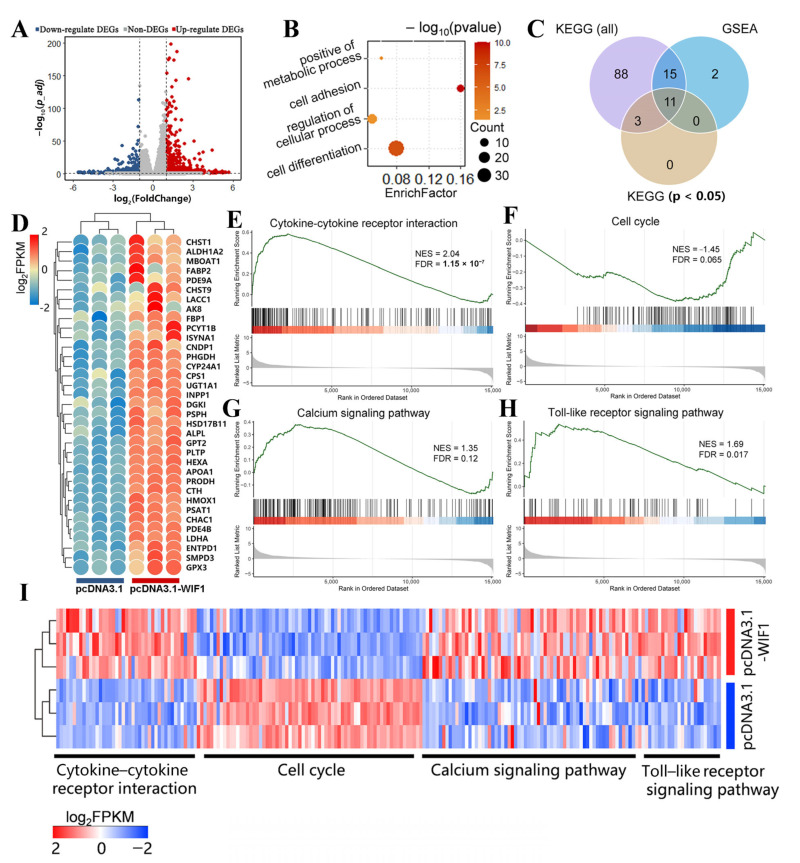
*WIF1* stimulates metabolic activity of chicken prehierarchal GCs (*n =* 3). (**A**) Volcano plot of differentially expressed genes (DEGs). The criteria for DEGs screening were |$\log_{2}\left(FoldChange\right)$| > 1 and *p*_adj < 0.05. (**B**) Gene Ontology (GO) terms enriched for DEGs. (**C**) Venn diagram of Kyoto Encyclopedia of Genes and Genomes (KEGG) enrichment analyses of DEGs and gene set enrichment analysis (GSEA) of all genes. KEGG (all) represents all pathway terms of KEGG analysis. KEGG (*p* < 0.05) represents the result of all results of KEGG analysis filtered with *p* < 0.05. GSEA represents the results filtered by |NES|>1, *p*_value < 0.05, and FDR < 25%. (**D**) Heatmap of metabolism-related genes. GSEA of significant pathways in *WIF1*-overexpressed and control pre-GCs: cytokine–cytokine receptor interaction pathway (**E**), cell cycle (**F**), calcium signaling pathway (**G**), and toll–like receptor signaling pathway (**H**). (**I**) Heatmap of the genes involved in the GSEA analysis.

**Figure 7 ijms-25-01788-f007:**
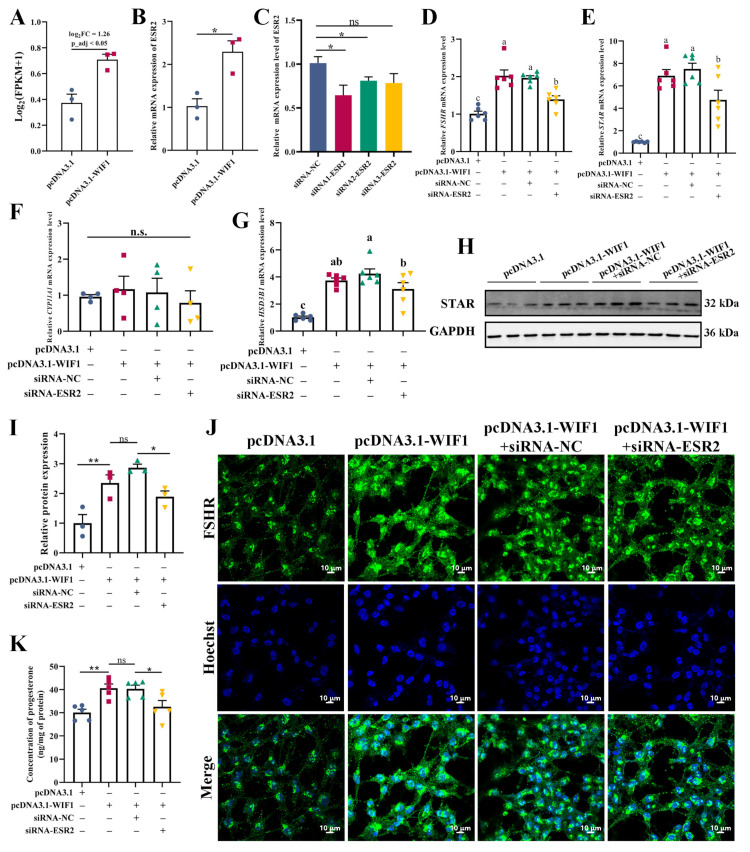
*WIF1* promotes progesterone synthesis through upregulation of *ESR2* in GCs. (**A**) Bar plot of *ESR2* in RNA-seq analysis (*n =* 3). (**B**) mRNA expression levels of *ESR2* (*n =* 3). (**C**) mRNA expression levels of *ESR2* after transfection with siRNA-ESR2 in GCs (*n =* 5). (**D**–**G**) mRNA expression levels of *FSHR* (*n =* 6), *STAR* (*n =* 6), *CYP11A1* (*n =* 4), and *HSD3B1* (*n =* 6). (**H**,**I**) Protein expression levels of STAR in GCs (*n =* 3). (**J**) Immunofluorescence staining of FSHR in chicken GCs, scale bar: 10 μm. (**K**) Concentration of progesterone in chicken GCs assessed by ELISA (*n =* 5). The intracellular total protein content was assessed to normalize the progesterone hormone content. Results are shown as the mean ± SEM. Different lowercase letters are used to indicate the level of significance of differences (*p* < 0.05); the same lowercase letters indicate non-significant differences (*p* > 0.05). * *p* < 0.05, ** *p* < 0.01, and ns *p* ≥ 0.05.

**Figure 8 ijms-25-01788-f008:**
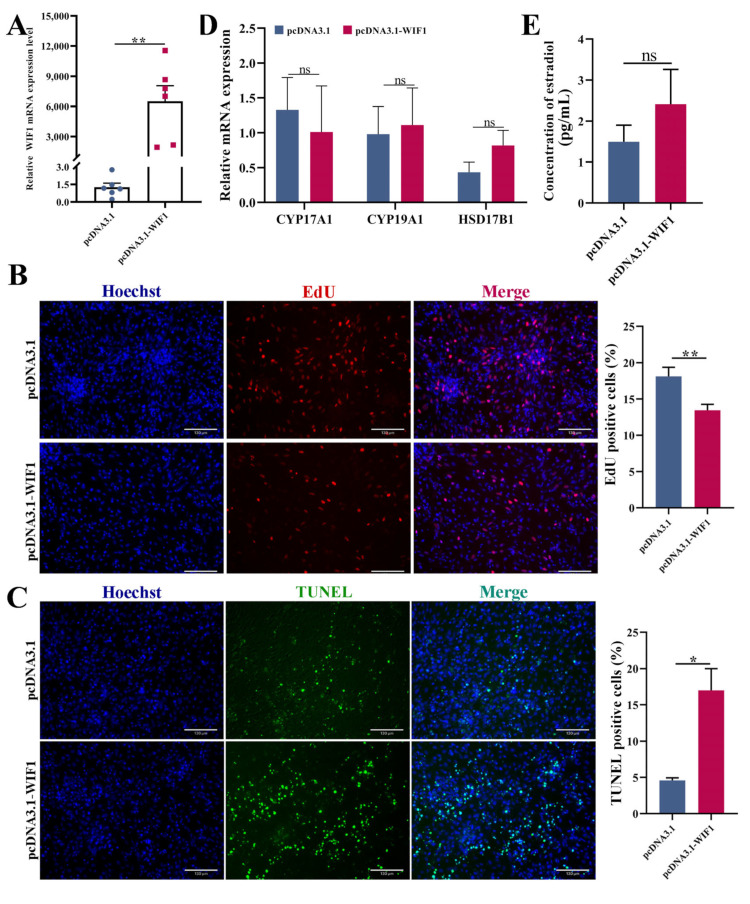
*WIF1* suppresses TC proliferation and promotes apoptosis. (**A**) mRNA expression levels of *WIF1* after transfection with pcDNA3.1-WIF1 in TCs (*n =* 6). (**B**) Proliferation of chicken TCs determined by 5-ethynyl-2′-deoxyuridine (EdU) after 24 h of transfection with pcDNA3.1-WIF1 (*n =* 7), scale bar: 130 μm. (**C**) The apoptosis of follicular TCs was detected by TdT-mediated dUTP Nick-End Labeling (TUNEL) after transfection with pcDNA3.1-WIF1 (*n =* 3), scale bar: 130 μm. (**D**) mRNA expression levels of *CYP17A1*, *CYP19A1*, and *HSD17B1* in TCs with *WIF1* overexpression (*n =* 3). (**E**) The concentration of estradiol in TCs with *WIF1* overexpression was assessed by ELISA. Results are shown as the mean ± SEM. * *p* < 0.05, ** *p* < 0.01, and ns *p* ≥ 0.05.

## Data Availability

The data presented in this study are available on request from the corresponding author.

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
