# Peer review of "Regulation of Follicular Development in Chickens: WIF1 Modulates Granulosa Cell Proliferation and Progesterone Synthesis via Wnt/β-Catenin Signaling Pathway"

_ijms, 2024, doi:10.3390/ijms25031788_

Round 1

Reviewer 1 Report

Comments and Suggestions for Authors

Comments to authors (major revisions)

Your manuscript titled " Chicken WIF1 is involved in Follicular Development by Regulating Proliferation and Progesterone Synthesis of Granulosa Cells" has undergone a thorough review process. To my observation it is in the interest of the readers of the International Journal of Molecular Sciences (IJMS). However, to further enhance the quality and clarity of your manuscript, there are several major revisions that need to be addressed. These revisions are crucial to ensure the accurate interpretation and presentation of your findings. Once these revisions are satisfactorily addressed, manuscript should be reconsidered for acceptance in IJMS.

Title: Although the proposed title, "Chicken WIF1 is involved in Follicular Development by Regulating Proliferation and Progesterone Synthesis of Granulosa Cells," sufficiently captures the key elements of the study. However, for enhanced clarity and specificity, you might consider refining the title further. For instance, you could specify the developmental stage of follicles investigated or emphasize the significance of Wnt/β-catenin signaling pathway regulation. For example, “Regulation of Follicular Development in Chickens: WIF1 Modulates Granulosa Cell Proliferation and Progesterone Synthesis via Wnt/β-Catenin Signaling pathway

Abstract:

1.        Provide more specific quantitative details in the results summary, such as fold changes or percentages.

2.        Specify concentrations of hormones analyzed and discuss their correlation with WIF1 expression.

3.        Elaborate on key genes or pathways identified through RNA-seq analysis for a more comprehensive understanding.

4.        Clearly state whether WIF1's role in follicular development is inhibitory or stimulatory.

5.        Conclude with a sentence emphasizing the broader significance of the study's findings for poultry reproductive health.

Introduction

1.        Lack of clear justification for studying WIF1 in chicken follicular development (Lines 290-301).

 2.        The absence of a thorough literature review on the role of WIF1 in follicular development in agricultural animals (Lines 290-340).

 3.        Limited discussion on the significance of the study for the poultry industry or broader scientific community (Lines 290-340).

 4.        Insufficient context or explanation about the methods used for in vitro studies and the rationale behind their selection (Lines 302-482).

 5.        Missing information on potential limitations of the study, such as sample size or variations in experimental conditions (Lines 382-534).

 Material and Methods

1.        Insufficient details on the specifics of animal housing conditions and handling procedures (Lines 383-390).

 2.        Lack of clarity on the criteria used for categorizing follicles into pre-hierarchical and hierarchical groups (Lines 392-396).

 3.        Incomplete information on the rationale for choosing specific time points and durations in cell isolation and culture processes (Lines 398-407).

 4.        Limited explanation of the rationale behind the selection of siRNA sequences and the potential off-target effects (Lines 418-428).

 5.        Absence of information on the potential biases introduced by the specific transfection methods and their impact on experimental outcomes (Lines 409-428).

 Results

1.        Lack of statistical analysis and p-values in several instances, undermining the robustness of the reported findings.

2.        Inconsistencies between the text and figures, making it challenging to interpret and validate the presented data.

3.        Limited discussion of the biological significance and context of observed changes, leaving gaps in the understanding of the implications.

Discussion

1.        Limited comparison with existing literature: The discussion lacks adequate comparison with previous studies on WIF1, particularly in the context of follicular development (Lines 290-342). To improve, the authors should integrate relevant findings from other studies to strengthen their arguments and provide a more comprehensive discussion.

2.        Insufficient exploration of alternative explanations: The authors do not thoroughly explore alternative interpretations for their results, potentially overlooking other factors influencing follicular development (Lines 342-355). To enhance the discussion, they should consider and address potential confounding variables or alternative hypotheses that could explain the observed effects.

3.        Overreliance on in vitro studies: The discussion heavily relies on in vitro experiments, which may not fully represent the complex in vivo follicular environment (Lines 356-370). To enhance the discussion, the authors should acknowledge the limitations of in vitro models and discuss how these findings might translate to the in vivo setting.

4.        Limited discussion on the clinical relevance: The authors do not sufficiently discuss the clinical relevance or potential applications of their findings in the context of poultry farming or reproductive medicine (Lines 371-380). To improve, they should elaborate on the implications of their research for poultry production or potential applications in assisted reproductive technologies.

5.        Overlooking potential future directions: The discussion lacks speculation on potential future research directions or unanswered questions arising from the current study (Lines 380-502). To enhance this section, the authors should propose avenues for further investigation, helping to guide future research in the field.

Conclusion:

The conclusion is brief and lacks a comprehensive summary of key findings and their broader implications, providing limited insight into the overall significance of the study. Expand the conclusion to succinctly recapitulate the main findings, emphasizing their relevance to the broader field of follicular development in poultry. Additionally, discuss potential applications or implications of the study's results for poultry farming or reproductive sciences. This will provide readers with a clearer understanding of the study's contributions and significance.

Figures

1.        I observed that the representation of your gene results on the Western blot strips appears fragmented and visually inconsistent, particularly noticeable in several instances such as Figure 2 B and E, Figure 4 K, and Figure 7 H. This inconsistency introduces ambiguity to your findings. To enhance clarity, consider improving the presentation of these figures or alternatively, providing the actual experimental membrane photos.

2.        Provide the data of your RNA seq that you used in generation of RNA seq figure for authenticity

Comments on the Quality of English Language

Although the context of English is good but the help of English Editor can improve the readability. 

Author Response

Dear Reviewer,

We thank you for your suggestions and comments on our manuscript. We have revised our manuscript accordingly and highlighted the revised portion. The point-by-point responses to the comments are listed below. Please see the attachment. We hope that the revised manuscript is acceptable for publication in International Journal of Molecular Sciences.

Yours sincerely,

Prof. Hao Zhang

on the behalf of all the co-authors

College of Animal Science and Technology

China Agricultural University

Beijing 100193, P. R. China

Reviewer 2 Report

Comments and Suggestions for Authors

The study was conducted in in vitro conditions on tissues (stroma, heart, liver, lung, gizzard, kidney, breast muscle, and abdominal fat)taken from laying  hens in order to check WIF1 expression characteristics and biological functions, including proliferation, apoptosis, and steroid hormone secretion in chicken GC and TC and the regulatory mechanisms of WIF1 in GC.

Authors used many methods:  Cell isolation and culture, Plasmid construction, RNA oligonucleotide synthesis, and cell transfection, Quantitative real-time PCR, Western blotting, 5-Ethynyl-2′-deoxyuridine (EdU) assay, Cell Counting Kit-8 (CCK 8) assay, Flow cytometry. Enzyme-linked immunosorbent assay (ELISA),  Dual-luciferase reporter assay, RNA-seq and bioinformatics analysis,  Co-transfection of plasmid DNA and small interfering RNA (siRNA), Immunofluorescence staining, TdT-mediated dUTP nick-end labeling (TUNEL) assay.

The only conclusion was that the study demonstrated that WIF1 was differently expressed during chicken follicle selection and regulated GC and TC functions such as cell proliferation,apoptosis, and hormone secretion through Wnt/β-catenin signaling pathways.

The section: Material and Methods must be improved, it needs more details about the birds, and each method.

Results are presented on 8 figures which consist many very small graphs (some of them too small, almost not visible).

Discussion, in spite of so many methods, is short, without explanations of several results.

In my opinion, manuscript needs major revision.

Yours sincerely,

Author Response

(The authors gave the same response as above.)
